# Ultrasound Evaluation of Upper Facial Muscles to Guide Botulinum Toxin Application

**DOI:** 10.3390/toxins17120595

**Published:** 2025-12-14

**Authors:** Dominika Jaguś, Anna Pawłowska, Robert Krzysztof Mlosek

**Affiliations:** 1Diagnostic Ultrasound Lab, Faculty of Medicine and Dentistry, Medical University of Warsaw, Kondratowicza 8, 03-242 Warsaw, Poland; 2Institute of Fundamental Technological Research, Polish Academy of Sciences, Pawińskiego 5B, 02-106 Warsaw, Poland

**Keywords:** botulinum toxin, upper facial muscles, ultrasound imaging, intra- and interindividual variability

## Abstract

Background: Botulinum toxin injection is one of the most common esthetic procedures, yet complications may occur due to anatomical variability or suboptimal injection technique. This study aimed to evaluate the upper facial muscles using ultrasound, focusing on inter- and intraindividual variability. Methods: The study involved volunteers aged 21–40 years, excluding those with prior facial treatments, trauma, or muscle disorders. The muscles examined included the occipitofrontalis (frontal belly), procerus, corrugator supercilii, and orbicularis oculi. Muscle thickness and distance from the epidermis were measured using high-frequency ultrasound. Statistical analyses included descriptive statistics, correlation with age and BMI, sex comparisons, and symmetry assessment. Results: A total of 127 participants (103 women and 24 men) were enrolled, with a mean age of 28.8 ± 4.4 years. Age showed no significant correlation with muscle thickness or depth, supporting the internal consistency of the studied age group. BMI showed moderate correlations with the depth of the selected forehead muscles. Males showed greater thickness in the frontal and procerus muscles. Relative side-to-side asymmetry coefficients reached 40% for both thickness and depth, indicating notable individual laterality. Conclusions: The study provides normative ultrasound parameters for the upper facial muscle in healthy adults. The results demonstrate significant anatomical variability depending on sex, BMI, and facial laterality, supporting individualized ultrasound-guided approaches for botulinum toxin injection.

## 1. Introduction

Botulinum toxin was first approved by the U.S. Food and Drug Administration (FDA) in 2002 for the treatment of glabellar lines. Since then, it has become one of the most frequently performed procedures aimed at facial rejuvenation [1,2]. Although the procedure has a favorable safety profile, it still involves a risk of complications as with any other medical treatment.

One of the most common adverse events is upper eyelid ptosis, occurring in approximately 0.59–5.4% of cases [3]. This complication is caused by toxin migration into the levator palpebrae superioris muscle, typically due to injections placed too close to the mid-pupillary line, high dosage, or excessive depth [4]. Another common complication is brow ptosis, occurring in fewer than 5% of cases, and characterized by a sense of forehead heaviness [5]. It results from excessive weakening of the frontalis muscle, especially when injections are placed too low or the dosage is too high [6]. Kroumpouzos et al. emphasize the importance of individualizing doses to avoid unintended effects [7].

Most complications can be prevented by understanding the functional anatomy of the upper face, using proper injection techniques, and performing ultrasound (US) evaluation of the injection site before treatment [4,8,9,10,11,12,13]. Additionally, many practitioners have shifted from standard injection points toward more individualized approaches that account for the patient’s unique facial features and esthetic goals. Advanced injection techniques enable esthetically pleasing eyebrow modeling while preserving natural facial expressions [14,15].

Although ultrasound is gaining importance in esthetic medicine, its use is still limited by a lack of practitioner training in facial anatomy imaging. Therefore, there is a need to promote knowledge in this area and to present the practical benefits of its application. Additionally, the scientific literature lacks clear guidelines regarding a standardized ultrasound examination protocol for facial muscles. Studies investigating the ultrasound assessment of the upper facial muscles show inconsistencies in methodology, exclude certain muscles from measurement and are based on cadaveric data [10,16,17,18,19].

This study aims to familiarize the reader with the possibilities of ultrasonographic evaluation of the upper facial muscles, focusing on interindividual and intraindividual anatomical variations. These differences may affect the risk of complications, and facial ultrasound is crucial in minimizing that risk [12,13]. Additionally, this study aims to provide unified and reproducible data on the morphological parameters of the upper facial muscles, which can provide a basis for further standardization of US examinations and individualized injection techniques.

## 2. Results

The final study cohort included 127 adult volunteers who met the study criteria, with no exclusion conditions reported. The group comprised 103 women and 24 men. Participants were aged 21 to 40 years, with a mean age of 28.8 ± 4.4 years (men: 28.6 ± 3.6; women: 28.8 ± 4.6).

### 2.1. Presentation of Ultrasonographic Anatomy

The examined structures are presented in Figure 1, Figure 2, Figure 3 and Figure 4, illustrating potential variability in their ultrasonographic appearance. All images were acquired with the probe positioned according to the study protocol.

### 2.2. Descriptive Statistics

A total of 14 dependent variables were obtained from the dataset, including muscle thickness and epidermis distance, measured across seven facial muscle regions. The results for these variables are presented in Table 1, including sample size, minimum and maximum values, mean, and standard deviation. The count values indicate the number of completed measurements for each muscle on both sides of the face, providing the exact number of measurements contributing to each variable. Differences between muscles reflect incomplete measurement sets for some individuals, resulting from partial data collection during the ultrasound examination.

### 2.3. Correlation Between Ultrasound Facial Measurements and Age/BMI

Spearman’s rank correlation coefficient was used to examine the dependence of the ultrasonographic parameters on age and BMI. The correlation results between age and muscle thickness, as well as their distance from the epidermis, are presented in Table 2. No statistically significant correlations were found for any of the examined ultrasonographic parameters and age, confirming the appropriateness of selecting the 21–40 years age group as a reference population.

The correlation results between BMI and muscle thickness and their distance from the epidermis are shown in Table 2. Only the thickness of mP and mOOcIP showed a statistically significant but weak correlation with BMI, with a correlation coefficient of 0.156 and 0.16, respectively. Moderate correlations between BMI and the depth were observed for the following muscles: mOF (0.273), mP (0.174), mCS (0.198), and mOOcIO (0.228).

### 2.4. Comparison of Ultrasound Facial Measurements Between Female and Male Groups

Ultrasonographic muscle parameters were grouped by sex. Due to violation of parametric test assumptions (unequal group sizes), nonparametric tests were used, including the median test for independent samples and the Mann–Whitney U test.

In Figure 5, violin plots represent the distribution of thickness measurements for each sex, supplemented by box plots marking the 25th, 50th (median), and 75th percentiles, along with the mean. Table 3 presents medians, *p*-values for both tests and effect sizes. Significant differences between sexes were observed in the thickness of forehead muscles (mOF, mP) and the upper parts of the orbicularis oculi muscle (mOOcSO, mOOcSP), with males exhibiting greater thickness. No significant differences were found for other muscles.

Muscle distances from the epidermis are shown using violin plots in Figure 6. Table 3 summarizes the median values and significance tests. A statistically significant difference was found for the mOOcIO muscle, where males had greater distances from the epidermis. Other muscles showed no significant sex-related differences.

### 2.5. Comparison of Ultrasound Facial Measurements Between the Right and Left Side

Ultrasonographic parameters of the right and left sides of the face were compared using paired-sample tests. Depending on normality assumptions, either Student’s *t*-test or the Wilcoxon signed-rank test was applied.

The distributions of asymmetry coefficients for muscle thickness are shown in Figure 7. Table 4 summarizes the comparative analysis of muscle thickness between the right and left sides. The strongest Pearson correlations (>0.6) were observed in the mOF and mP muscles. The asymmetry coefficient for muscles mOF, mP, mCS, mOOcSO, mOOcSP, mOOcIO, and mOOcIP reached up to 50%. No significant directional asymmetry was found in muscle thickness measurements.

Figure 8 shows the distributions of the asymmetry coefficient values for muscle distances from the epidermis. Results comparing muscle-to-epidermis distances on the right and left sides are presented in Table 4. The highest Pearson correlation coefficient (>0.6) was found for the depth of mP. The largest asymmetry coefficients were observed in the orbicularis oculi muscle group, reaching 50–70%. Dependent sample tests revealed statistically significant differences between the right and left side measurements for depth of mOF, mCS, and mOOcSP, with small-to-medium effect sizes (d = 0.29, 0.22, and 0.27, respectively). For the depth of mCS, a negative skewness (−0.163) indicates dominance of the left side. In contrast, the depth of mOF and mOOcSP showed positive skewness, negative median, and asymmetry coefficient distribution favoring the right side.

## 3. Discussion

This study proposes normative ultrasound values for the thickness and depth of upper facial muscles in healthy adults aged 21–40 years. The results suggest that muscle measurements may vary significantly depending on sex, BMI, and facial side. Increased BMI can be associated with greater muscle depth, particularly in the forehead region. Men may demonstrate thicker occipitofrontalis (frontal belly), procerus and upper eyelid orbicularis oculi muscles than women. Additionally, notable asymmetries can be found between the right and left sides of the face, with differences in both thickness and depth reaching up to 40%. These findings indicate that facial muscle anatomy is highly variable and influenced by individual characteristics.

A comparison between the current findings and those reported in previous studies is presented in Table 5. The frontal belly of the occipitofrontalis muscle originates in the skin of the eyebrows and glabella, then continues upward into the galea aponeurotica [20]. The fact that the muscle originates from the skin contributes to large differences in reported thickness values. Apart from the thin belly, numerous muscle fibers penetrate through the fatty tissue of the forehead area [20]. In dynamic ultrasound studies, i.e., during contraction and relaxation of the mOF, the activity of both the frontal belly and muscle fibers within the mentioned fatty tissue is visible. Consequently, the literature shows different methods of defining the frontal belly of the mOF. One method is the measurement from the lower edge of the muscle to the subcutaneous layer, which includes muscle fibers together with the fatty tissue. This method was used in publications [16,17,21]. Another method, presented in this work, is the measurement of the main muscle mass, excluding fibers within the fatty tissue. A similar mOF image, without thickness assessment, was presented by Wu et al. [10]. In this study, the chosen measurement approach assumed that fat layer thickness rises with increasing obesity, which would interfere with assessing muscle thickness in relation to factors other than BMI. The mOF boundaries are often difficult to distinguish, especially in individuals with overweight or obesity, which was confirmed in this study and by Alfen et al. [22]. Moreover, using ultrasound probes operating below 15 MHz also complicates or even prevents clear identification of the main muscle mass of the belly, as shown in the studies by Volk et al. [16,17]. The discrepancies in mean muscle thickness between the obtained results and those reported in the cited studies may be due to the discussed differences.

A difference in the thickness of mP was observed (see Table 5) between the results of this study and those reported by Alfen et al. [22]. A possible cause of the discrepancy is the small sample size in their study (12 participants). Additionally, their measurements were taken in the area of the glabella, which is located above the expected main mass of the procerus muscle.

The mCS thickness measurements shown in Table 5 are derived from cadaveric studies, where muscle width was assessed in the coronal plane. This methodological difference likely contributes to the observed discrepancies. Cadaver-based findings were used as a reference point due to the lack of ultrasound-derived measurements.

The measurements of the orbital part thickness (mOOcSO) and depth (mOOcIO) align with values reported in the literature [16,17,24], confirming that ultrasound enables clear and consistent identification of the orbital part of the orbicularis oculi muscle (Table 5). This suggests that even less experienced operators can accurately identify this region, facilitating well-targeted botulinum toxin administration. It is especially relevant given that the orbital part is the primary target site for toxin deposition [8]. Given the lack of published data on mOOcSP and mOOcIP, this study may contribute valuable baseline data for further investigations.

An increase in BMI, as an indicator of obesity, was associated with a greater distance from the epidermis in all examined facial muscles except the orbicularis oculi. These findings are supported by Thiemann et al., who reported medium effect sizes between BMI and soft tissue depth at the metopion, supraglabella, and glabella landmarks, based on multi-slice computed tomography analysis [25]. Based on these data, forehead muscles are expected to be located deeper in obese individuals.

The thickness of the forehead muscles (mOF, mP) and the upper parts of the orbicularis oculi muscle (mOOcSO, mOOcSP) was greater in men than in women. Similar results were reported by Volk et al., who found significantly thicker orbicularis oculi muscles in men compared to women [16]. However, in their study, the mOF was statistically thicker in women than in men, which may be due to a different measurement method that included muscle fibers within the fatty tissue layer. Among other authors cited in the discussion of descriptive statistics, sex-related differences were not statistically significant. Regarding muscle depth, one part of the orbicularis oculi muscle (mOOcIO) was found to be significantly deeper in men than in women. In contrast, Hormazabal-Peralta et al. found the orbicularis oculi muscle to be deeper in women [24]. This variation may reflect ethnic differences in facial anatomy, particularly in fat pad distribution, as their study population consisted of Korean individuals.

Asymmetry in muscle thickness was observed across all examined facial muscles. In some individuals, side-to-side differences reached up to 50%, meaning that one muscle was up to 50% thicker than its contralateral counterpart. Pinar et al. similarly reported high rates of asymmetry in the corrugator supercilii, with 45% of subjects showing complete asymmetry and 25% partial symmetry [18]. The mOOcSP muscle showed the greatest thickness asymmetry, with a quartile deviation of approximately 22% and a distribution skewed toward greater thickness on the right side, although this right-side dominance was not statistically confirmed. This observation requires a larger sample size.

Asymmetry in muscle depth relative to the epidermis was apparent across all analyzed facial muscles. For some muscles, like mOOcSP, differences reached up to 70%, indicating considerable side-to-side variability. These findings suggest that even small facial muscles can differ substantially between sides. A tendency toward right-sided dominance was observed for mOF and mOOcSP, while mCS was deeper on the left. However, the underlying causes and consistency of these asymmetry patterns require further investigation.

The anatomical variability observed in this study may offer preliminary insights into the clinical context of upper facial botulinum toxin injections. The consistently greater muscle thickness in men compared with women, particularly in the procerus, frontalis, and specific portions of the orbicularis oculi, may partly explain sex-related differences in responsiveness or required dosing reported in clinical practice [26,27]. A recent review on neuromodulator use in men reported that male patients typically require higher doses, reflecting greater muscle mass and distinct facial anatomical characteristics compared with women [26]. Similarly, a cross-sectional analysis of treatment patterns demonstrated that men received significantly greater units of botulinum toxin in the procerus and nasal muscles than women [27]. These findings suggest that the sex-related differences in muscle thickness and depth identified in our study may help explain clinically relevant variability in responsiveness and dosing requirements. Additionally, measurable side-to-side differences in several upper facial muscles suggest that injection doses may need to be adjusted individually for each side. Even small anatomical variations can influence local diffusion and treatment outcomes. This concept is supported by in silico modeling [28], which demonstrated that minor changes in injection volume or placement can significantly affect botulinum toxin diffusion, highlighting the potential clinical relevance of subtle inter-side anatomical differences. These observations imply that clinical observations correspond to measurable variations in ultrasound-derived data, emphasizing their relevance for guiding more personalized treatment strategies.

Age also contributes importantly to the anatomical patterns observed in clinical settings. Although limited to young adults, our findings must be interpreted in the context of well-established, age-related changes affecting the upper facial muscles. Aging of the upper facial muscles and overlying skin, including the frontalis, corrugator, and procerus, is associated with thinning, reduced muscle activity, and decreased tissue elasticity, as shown by anatomical, ultrasound, and functional studies [29,30,31]. These processes contribute to greater inter-individual variability in both muscle thickness and depth from the epidermis. Consequently, in older patients seeking forehead and periocular rejuvenation, pre-treatment ultrasound imaging is expected to offer even greater value by enabling more precise, individualized injection planning and further reducing complications such as brow or eyelid ptosis.

Recent evidence highlights the clinical advantages of ultrasound-guided injections over traditional landmark-based approaches. Systematic review data indicates that ultrasound guidance improves anatomical precision, enhances safety, and may optimize treatment outcomes across a variety of facial muscles innervated by the facial nerve [32]. In addition, controlled studies using cadaveric models and body donors demonstrate that ultrasound guidance allows more accurate targeting of specific muscles, reducing the risk of inadvertent diffusion and associated complications [33]. These findings support the potential clinical benefit of incorporating ultrasound into routine facial injection practice.

In practice, ultrasound guidance may be implemented either in real time or as a tool to pre-identify the injection sites. Both approaches increase the total duration of the visit compared with standard injections. However, the perceived benefit of this additional time is subjective and depends on the practitioner’s experience and workflow. It can be justified by the potential for improved precision in targeting muscle bellies and optimized dosing. Successful implementation of ultrasound-guided procedures requires the injector to have appropriate training and experience in facial ultrasound. In particular, clinicians’ existing knowledge of facial anatomy should be expanded to include the ultrasound appearance of the relevant muscles, ensuring accurate identification and assessment during the procedure.

A similar need for detailed anatomical knowledge is reflected in cadaveric research illustrating notable differences in upper-face muscle morphology across individuals. For the frontalis muscle, several distinct morphological variants differing in fiber arrangement and interdigitation patterns have been described [34]. The corrugator supercilii likewise presents multiple anatomical trajectories and attachment patterns, as demonstrated in a cadaveric analysis linking these variations to differences in injection technique [35]. Moreover, detailed dissection of the superior orbital region has shown that the muscular–deep fascial system around the orbit is structurally heterogeneous, affecting the configuration of adjacent periorbital muscles such as the orbicularis oculi [36]. The variability reported in these anatomical studies is consistent with the heterogeneity observed in our ultrasound measurements and supports the usefulness of pre-procedural ultrasonography to account for individual differences during injection planning.

A limitation of this study is the unequal gender distribution, with a significant predominance of female participants. This imbalance resulted from difficulties in recruiting male participants for facial ultrasound and frequent withdrawal from the procedure, often caused by reported discomfort, especially following gel application near the eyes. The uneven sex distribution limited the ability to detect small sex-related differences. However, post hoc evaluation indicated that the study had sufficient power (>80%) to detect medium-to-large effects, which were observed for several muscles. Another limitation was the sample size, which did not allow for the use of more advanced statistical analyses, including nonlinear models. Additionally, the study population was homogeneous in terms of ethnicity, which may limit the generalizability of the findings to more diverse populations. In future studies, it would be valuable to include a more ethnically diverse population and a broader age range to better reflect anatomical variation across the human lifespan. A broader age range would also enable a more detailed analysis of age-related structural changes, such as alterations in echogenicity, tissue elasticity or muscle architecture, which cannot be captured in a young healthy cohort and may provide additional parameters relevant for future studies. Within healthy populations, future investigations could enhance geometric muscle characterization to identify structural parameters that could subsequently be linked to clinical outcomes. In addition, establishing minimum clinically important difference values for ultrasound-based assessments would allow future studies to perform targeted hypothesis testing and design adequately powered clinical trials.

This study demonstrates that facial anatomy can be influenced by BMI, sex, and even by side-to-side variation within the same person. The risk of complications from botulinum toxin injections is influenced not only by injection technique and dosage but also by the unique anatomy of each patient. Many studies have focused on avoiding complications like eyelid or brow ptosis and double vision through the identification of “safe zones” [7,11,15]. For instance, a recommended technique to avoid side effects includes injecting at least 1 cm above the orbital rim in the glabellar area, as well as aiming for the middle and upper parts of the frontalis muscle in forehead treatments [11]. However, given the high variability in the topography of anatomical structures, relying on arbitrarily defined points may be insufficient. Ultrasound muscle examination may allow for precise identification of muscle origins and insertions, as well as assessment of thickness and depth. These details are important for the effective planning of neuromodulator procedures.

## 4. Conclusions

Ultrasonography is an effective method for assessing facial anatomical structures. It enables precise measurement of muscle thickness and the distance between muscles and the epidermis. Facial anatomy demonstrates substantial variability related to BMI, sex, and laterality within individuals. As a diagnostic tool, ultrasonography can support the identification of botulinum toxin injection sites. It may also serve as an intra-procedural monitoring technique aimed at minimizing the risk of post-procedural complications.

## 5. Materials and Methods

The study was conducted using high-frequency ultrasonographic diagnostics of four upper facial muscles. Ultrasound examinations were performed between 2021 and 2022 at the Ultrasound Diagnostics Lab, Department of Pediatric Radiology, Faculty of Medicine and Dentistry, Medical University of Warsaw. The study was conducted in accordance with the Declaration of Helsinki and was approved by the institutional review board. All participants provided written informed consent for participation.

### 5.1. Study Participants

A total of 127 adult participants, aged 21 to 40 years and representing both sexes, were enrolled in the study. All participants volunteered specifically for this study and were recruited through open online announcements. For each participant, gender, age, and self-reported height and weight were recorded. This specific age group is widely used in the literature as a reference population for assessing normative values of facial muscles [37,38,39]. As the study was exploratory and descriptive, no formal a priori sample size calculation was warranted.

Exclusion criteria for study participation were a history of esthetic procedures affecting the structure or arrangement of muscles, including facial implants, fillers, botulinum toxin injections; history of facial trauma resulting in a fracture of facial bones, jaws or injury requiring the intervention of a plastic surgeon; facial nerve palsy; muscular dystrophies; myasthenia; and claustrophobia.

### 5.2. Ultrasound Imaging

Ultrasound examinations were performed using the Samsung RS80 system with a linear transducer LA4-18B (Samsung Medison, Seoul, Republic of Korea) operating within a frequency range of 4–18 MHz. Both halves of the upper face were examined in each volunteer. Minimal tissue compression was applied during the examination. Throughout all examinations, the acoustic safety indices—Mechanical Index (MI) and Thermal Index (TI)—remained within the range recommended by the British Medical Ultrasound Society [40].

The muscles examined included: frontal belly of the occipitofrontalis muscle (mOF); procerus muscle (mP); corrugator supercilii muscle (mCS); orbicularis oculi muscle, superior eyelid orbital part (mOOcSO); orbicularis oculi muscle, superior eyelid palpebral part (mOOcSP); orbicularis oculi muscle, inferior eyelid orbital part (mOOcIO); and orbicularis oculi muscle, inferior eyelid palpebral part (mOOcIP). These four muscles are commonly targeted in botulinum toxin treatments for upper face rejuvenation.

Patients were examined in a supine position, lying comfortably with the head positioned in line with the spine and facing straight ahead. Muscles were imaged in a relaxed state. Minimal pressure was applied to the tissues. The transducer was placed gently on the skin with a gel, without applying additional pressure from the probe’s weight or the operator’s hand.

Muscle imaging was performed according to the following protocol:Frontal belly of the occipitofrontalis (mOF): The transducer was positioned longitudinally lateral to the supraorbital foramen, with measurements taken along the convergence line.Procerus (mP): The transducer was positioned transversely over the nasal bone, the measurement was taken at half the muscle length.Corrugator supercilii (mCS): The transducer was positioned obliquely on the nasal part of the frontal bone. Placement was adjusted individually to identify the thickest section of the muscle. Measurements were obtained at half the muscle length.Orbicularis oculi (mOO): The transducer was placed longitudinally at the midpoint of the imaginary line connecting the medial and lateral canthi, with the transducer’s edges resting on the superior and inferior orbital rims.superior eyelid orbital component (cSO): measurement was taken at the superior orbital rim (bony edge).superior eyelid palpebral component (cSP): measurement was taken within 1 cm from the muscle margin.inferior eyelid orbital component (cIO): measurement was taken at the inferior orbital rim (bony edge).inferior eyelid palpebral component (cIP): measurement was taken within 1 cm from the muscle margin.


The method for measuring muscle thickness and its distance from the epidermis, along with probe positioning, is illustrated in Figure 9. Measurements were repeated three times for each muscle, and the mean value was used for analysis to increase measurement precision. All measurements were acquired by a single experienced operator, with 5 years of experience in facial ultrasound and 10 years of experience in diagnostic ultrasound overall. After data collection, all images and measurements were independently reviewed by a second expert, who had 30 years of experience in facial ultrasound and 35 years of experience in ultrasound in general. No discrepancies or errors requiring re-evaluation were identified, ensuring consistency and adherence to the protocol.

### 5.3. Statistical Analysis

Statistical analysis was preceded by exploratory examinations to ensure data integrity [41,42]. Basic descriptive statistics—including sample size, minimum and maximum values, mean, and standard deviation—were then calculated for muscle parameters (thickness and muscle-to-epidermis distance), allowing comparison with values reported in the literature.

Subsequently, relationships between ultrasonographic parameters and patients’ age and body mass index (BMI) were assessed using Spearman’s rank correlation coefficient due to non-normal data distribution and outliers in some of the analyzed variables [43]. Due to the significant correlation between weight and height, these variables were combined into BMI for analysis.

Next, male and female participants were compared using violin plots, the median test for independent samples, and the Mann–Whitney U test. Violin plots visualized data distribution, outliers, and central tendencies, while the median and Mann–Whitney tests evaluated differences in central values and distributions between the groups [44,45]. Additionally, effect sizes for the Mann–Whitney U test were calculated using the coefficient r, derived from the standardized test statistic, to quantify the magnitude of between-group differences [46].

The final analysis compared the right and left sides of the face using methods for dependent samples. Pearson’s correlation coefficient was used to assess the degree of symmetry, where values close to 1 indicate high similarity between the left and right sides (ideal symmetry: L = R). Additionally, an asymmetry coefficient was calculated according to the formula [47]:(1)$$Asymmetry\;coefficient=\frac{L-R}{max(L,R)}\times 100\%$$
The asymmetry coefficient distributions were characterized by descriptive statistics (skewness, kurtosis, median, and quartile deviation) [48]. Dependent sample comparisons were performed using the paired-samples Student’s *t*-test and, as a nonparametric alternative, the Wilcoxon signed-rank test [49]. For outcomes showing statistically significant side-to-side differences, effect sizes were calculated using Cohen’s d for paired samples to quantify the magnitude of these effects [46].

All statistical analyses were conducted using STATISTICA 10 (StatSoft Inc., Tulsa, OK, USA), with *p* < 0.05 considered statistically significant.

## Figures and Tables

**Figure 1 toxins-17-00595-f001:**
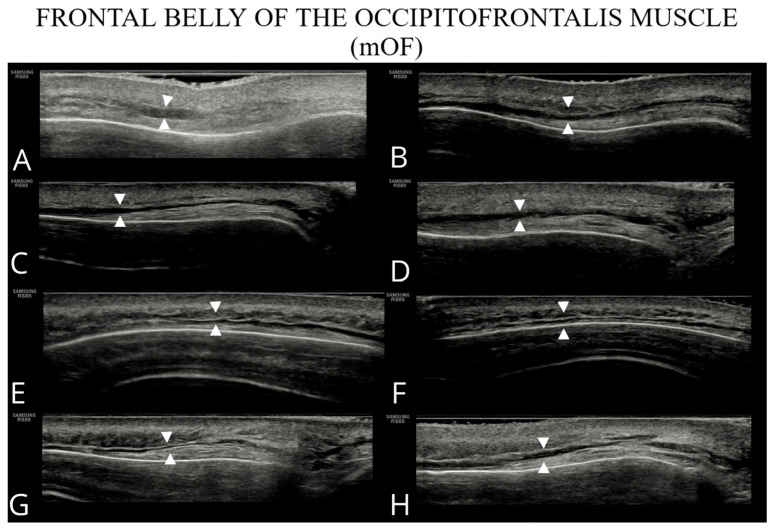
Ultrasound images of the frontal belly of the occipitofrontalis (**A**–**H**).

**Figure 2 toxins-17-00595-f002:**
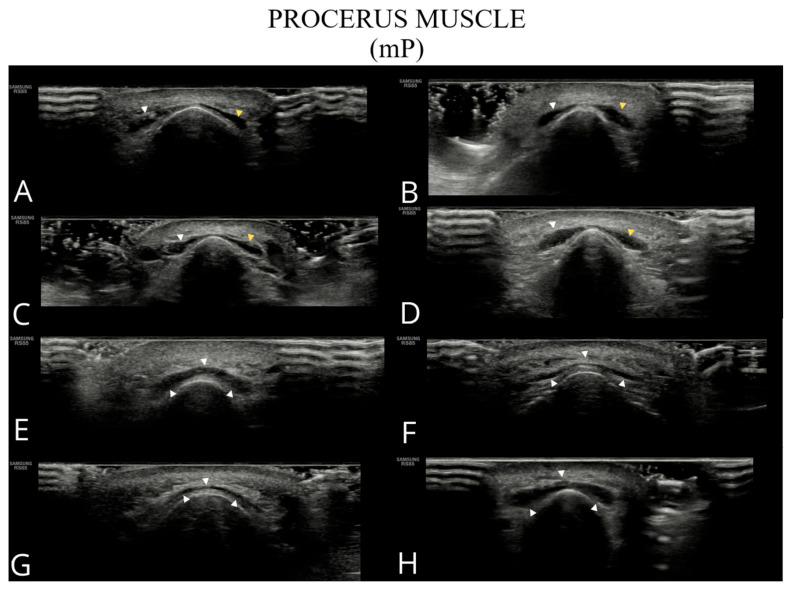
Ultrasound images of the procerus muscle (**A**–**H**). Panels (**A**–**D**) show a typical bilateral procerus (right and left bellies), while panels (**E**–**H**) present an anatomical variant with a single belly.

**Figure 3 toxins-17-00595-f003:**
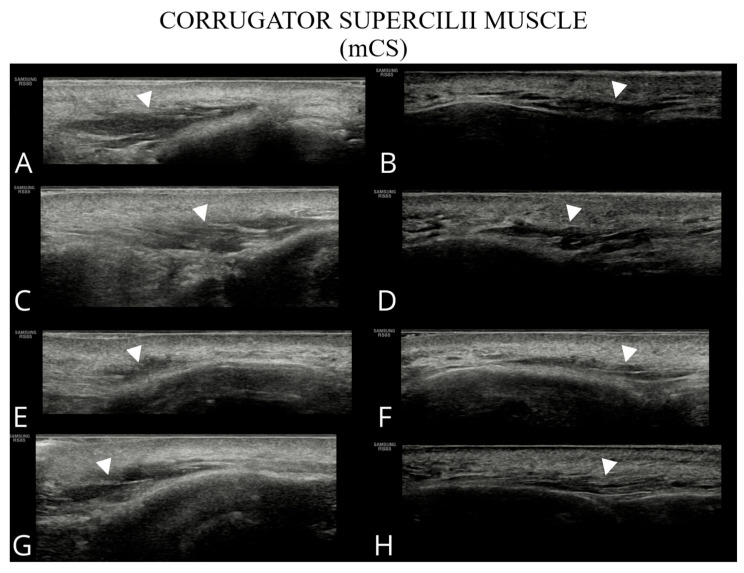
Ultrasound images of the corrugator supercilii muscle (**A**–**H**). Images (**A**,**C**,**E**,**G**) correspond to the right side of the patient, whereas (**B**,**D**,**F**,**H**) show the left side.

**Figure 4 toxins-17-00595-f004:**
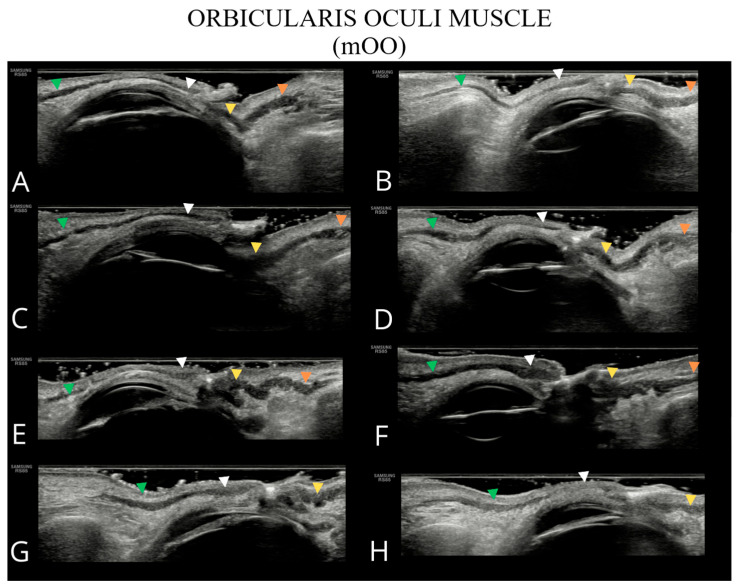
Ultrasound images of the orbicularis oculi muscle (**A**–**H**). The following parts of the orbicularis oculi muscle are indicated: orbital part of the upper eyelid—green arrowhead; palpebral part of the upper eyelid—white arrowhead; palpebral part of the lower eyelid—yellow arrowhead; and orbital part of the lower eyelid—orange arrowhead. In panels (**G**,**H**), the inferior orbital part of the orbicularis oculi muscle is not visualized due to the absence of a well-defined bony margin of the orbit.

**Figure 5 toxins-17-00595-f005:**
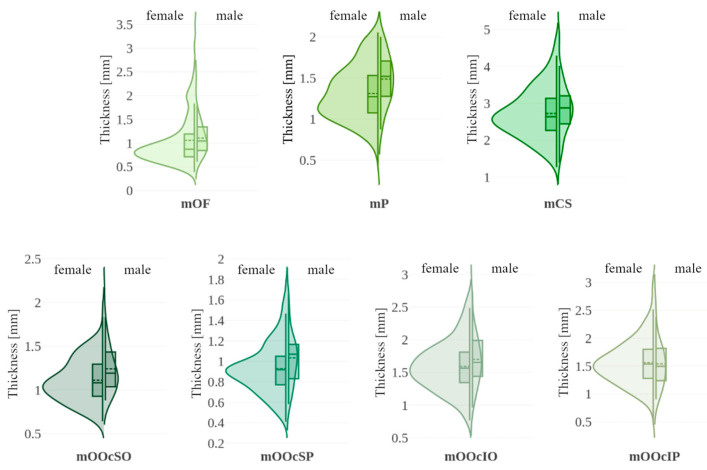
Violin plots illustrating muscle thickness with data stratified by sex (female vs. male). Each violin includes a boxplot (solid line: 25th, 50th [median], and 75th percentiles with whiskers) and the mean value (dashed line). mOF—frontal belly of the occipitofrontalis muscle; mP—procerus muscle; mCS—corrugator supercilii muscle; mOOcSO—orbicularis oculi muscle, superior eyelid orbital part; mOOcSP—orbicularis oculi muscle, superior eyelid palpebral part; mOOcIO—orbicularis oculi muscle, inferior eyelid orbital part; mOOcIP—orbicularis oculi muscle, inferior eyelid palpebral part.

**Figure 6 toxins-17-00595-f006:**
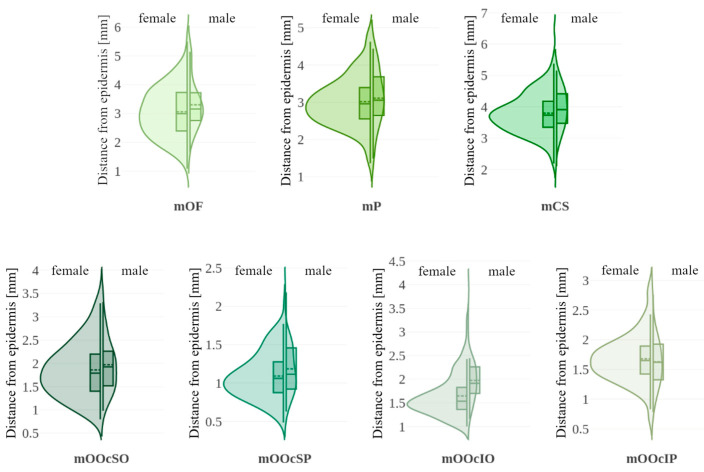
Violin plots illustrating muscle distance from the epidermis with data stratified by sex (female vs. male). Each violin includes a boxplot (solid line: 25th, 50th [median], and 75th percentiles with whiskers) and the mean value (dashed line). mOF—frontal belly of the occipitofrontalis muscle; mP—procerus muscle; mCS—corrugator supercilii muscle; mOOcSO—orbicularis oculi muscle, superior eyelid orbital part; mOOcSP—orbicularis oculi muscle, superior eyelid palpebral part; mOOcIO—orbicularis oculi muscle, inferior eyelid orbital part; mOOcIP—orbicularis oculi muscle, inferior eyelid palpebral part.

**Figure 7 toxins-17-00595-f007:**
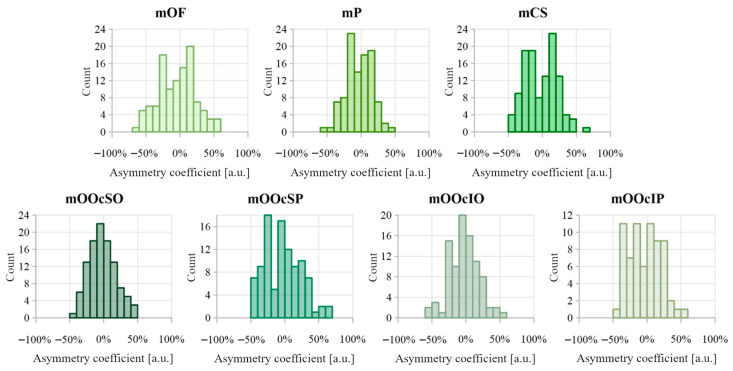
Distribution of asymmetry index values calculated between muscle thickness on the right and left side of the face. Negative asymmetry coefficient values indicate that the thicker muscle was on the right side, while positive values indicate that the thicker muscle was on the left side.

**Figure 8 toxins-17-00595-f008:**
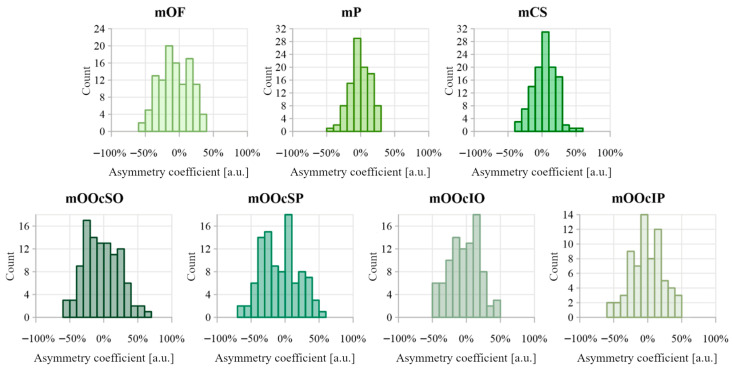
Distribution of asymmetry index values calculated between distance from the epidermis for muscles on the right and left sides of the face. Negative asymmetry coefficient values indicate that the deeper muscles are on the right side, while positive values indicate that the deeper muscles are on the left side.

**Figure 9 toxins-17-00595-f009:**
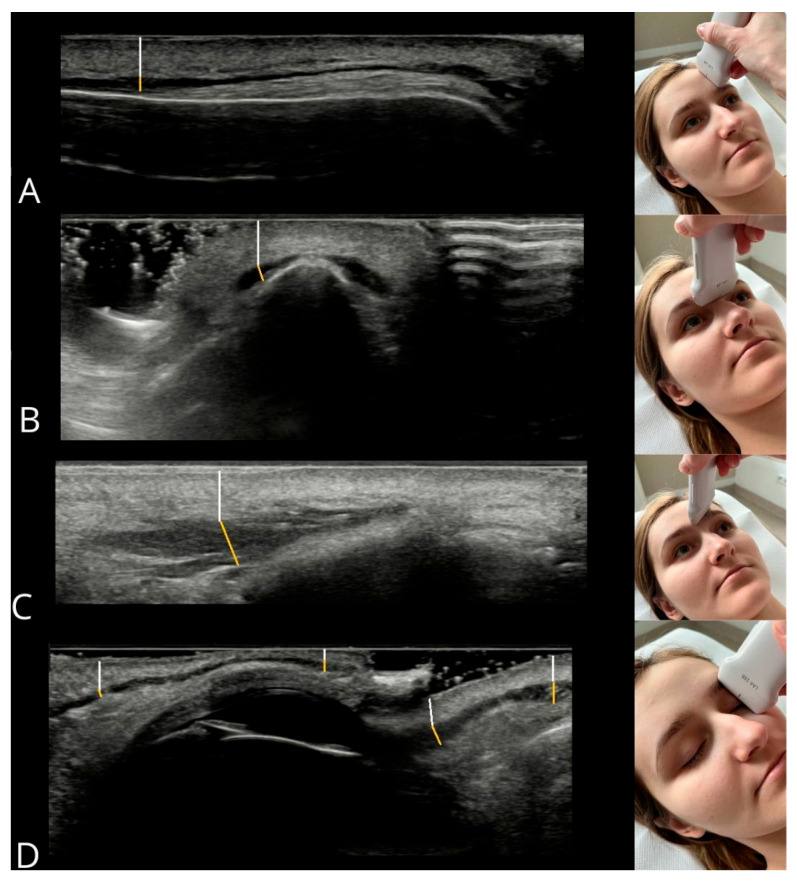
Measurement method for muscle thickness (yellow line) and epidermis-to-muscle distance (white line). Muscles: (**A**) frontal belly of occipitofrontalis, (**B**) procerus, (**C**) corrugator supercilii, (**D**) orbicularis oculi components (from left: upper eyelid—orbital, palpebral; lower eyelid—palpebral, orbital).

**Table 1 toxins-17-00595-t001:** Descriptive statistics (sample size, minimum, maximum, mean, standard deviation) for muscle thickness and muscle-to-epidermis distance.

		Count	Minimum [mm]	Maximum [mm]	Mean [mm]	Standard Deviation [mm]
**Thickness**	**mOF**	238	0.387	3.497	1.059	0.517
	**mP**	207	0.449	2.056	1.345	0.328
	**mCS**	243	1.268	4.920	2.741	0.666
	**mOOcSO**	231	0.642	2.128	1.138	0.260
	**mOOcSP**	220	0.410	1.766	0.948	0.251
	**mOOcIO**	210	0.765	2.727	1.615	0.374
	**mOOcIP**	181	0.521	3.015	1.563	0.413
**Distance from epidermis**	**mOF**	238	1.087	5.779	3.105	0.882
	**mP**	207	1.361	4.876	3.019	0.664
	**mCS**	243	2.107	6.484	3.824	0.660
	**mOOcSO**	231	0.795	3.578	1.878	0.583
	**mOOcSP**	220	0.485	2.183	1.101	0.302
	**mOOcIO**	210	1.002	3.897	1.692	0.462
	**mOOcIP**	181	0.667	2.942	1.668	0.407

mOF—frontal belly of the occipitofrontalis muscle; mP—procerus muscle; mCS—corrugator supercilii muscle; mOOcSO—orbicularis oculi muscle, superior eyelid orbital part; mOOcSP—orbicularis oculi muscle, superior eyelid palpebral part; mOOcIO—orbicularis oculi muscle, inferior eyelid orbital part; mOOcIP—orbicularis oculi muscle, inferior eyelid palpebral part.

**Table 2 toxins-17-00595-t002:** Summary of Spearman correlation coefficients calculated for the relationships between age and BMI with muscle thickness and distance from the epidermis. Statistically significant coefficients are highlighted in bold and underlined.

	Spearman Correlation Coefficient [a.u.] (*p*-Value [a.u.])			
	Age		BMI	
	Thickness	Distance from Epidermis	Thickness	Distance from Epidermis
**mOF**	0.090 (0.167)	−0.080 (0.216)	0.111 (0.090)	** 0.273 (<0.001) **
**mP**	0.113 (0.105)	−0.050 (0.477)	** 0.156 (0.025) **	** 0.174 (0.013) **
**mCS**	0.099 (0.126)	0.009 (0.892)	0.035 (0.588)	** 0.198 (0.002) **
**mOOcSO**	−0.001 (0.985)	0.032 (0.626)	0.009 (0.891)	0.025 (0.708)
**mOOcSP**	0.052 (0.445)	0.034 (0.614)	0.088 (0.192)	−0.017 (0.802)
**mOOcIO**	0.077 (0.264)	0.023 (0.744)	0.051 (0.460)	** 0.228 (0.001) **
**mOOcIP**	−0.015 (0.843)	−0.105 (0.161)	** 0.160 (0.031) **	0.077 (0.306)

BMI—body mass index; mOF—frontal belly of the occipitofrontalis muscle; mP—procerus muscle; mCS—corrugator supercilii muscle; mOOcSO—orbicularis oculi muscle, superior eyelid orbital part; mOOcSP—orbicularis oculi muscle, superior eyelid palpebral part; mOOcIO—orbicularis oculi muscle, inferior eyelid orbital part; mOOcIP—orbicularis oculi muscle, inferior eyelid palpebral part.

**Table 3 toxins-17-00595-t003:** Comparative results for female and male groups regarding muscle thickness and distance from the epidermis. The table includes medians for both groups, statistical significance values obtained from the independent-samples median test and the Mann–Whitney U test along with effect sizes. Statistically significant values are highlighted in bold and underlined.

		Median [mm]		*p*-Value [a.u.]		Effect Size [a.u.]
		Females	Males	Median Test	Mann–Whitney U Test	
**Thickness**	**mOF**	0.870	1.044	** 0.012 **	** 0.025 **	0.145
	**mP**	1.273	1.521	** 0.004 **	** 0.001 **	0.222
	**mCS**	2.638	2.872	0.095	0.152	0.092
	**mOOcSO**	1.081	1.191	** 0.028 **	** 0.007 **	0.179
	**mOOcSP**	0.918	1.068	** 0.036 **	** 0.017 **	0.161
	**mOOcIO**	1.564	1.653	0.134	0.146	0.100
	**mOOcIP**	1.538	1.496	0.722	0.717	0.027
**Distance from epidermis**	**mOF**	2.990	3.156	0.400	0.148	0.094
	**mP**	2.957	3.053	0.460	0.310	0.071
	**mCS**	3.732	3.911	0.095	0.165	0.089
	**mOOcSO**	1.790	1.926	0.388	0.265	0.073
	**mOOcSP**	1.060	1.115	0.294	0.120	0.105
	**mOOcIO**	1.534	1.912	** <0.001 **	** <0.001 **	0.280
	**mOOcIP**	1.650	1.622	0.456	0.521	0.048

mOF—frontal belly of the occipitofrontalis muscle; mP—procerus muscle; mCS—corrugator supercilii muscle; mOOcSO—orbicularis oculi muscle, superior eyelid orbital part; mOOcSP—orbicularis oculi muscle, superior eyelid palpebral part; mOOcIO—orbicularis oculi muscle, inferior eyelid orbital part; mOOcIP—orbicularis oculi muscle, inferior eyelid palpebral part.

**Table 4 toxins-17-00595-t004:** Comparative results for the right and left sides of the face regarding muscle thickness and distance from the epidermis. The table includes Pearson correlation coefficients, descriptive statistics of the asymmetry coefficient distribution, and statistical significance values obtained from tests for dependent samples. Statistically significant values are highlighted in bold and underlined.

		Pearson Correlation Coefficient [a.u.]	Descriptive Statistics for the Distribution of the Asymmetry Coefficient				*p*-Values for Tests of Dependent Samples [a.u.]
			Skewness [a.u.]	Kurtosis [a.u.]	Median [a.u.]	Quartile Deviation [a.u.]	
**Thickness**	**mOF**	** 0.712 **	−0.056	−0.421	−3.76%	18.55%	0.337 (tW)
	**mP**	** 0.617 **	−0.030	−0.321	−3.90%	14.27%	0.092 (ttS)
	**mCS**	** 0.374 **	0.137	−0.761	−0.92%	19.03%	0.358 (ttS)
	**mOOcSO**	** 0.404 **	0.325	−0.297	−4.03%	13.42%	0.267 (tW)
	**mOOcSP**	0.169	0.366	−0.509	−1.90%	21.93%	0.252 (tW)
	**mOOcIO**	** 0.313 **	0.216	0.301	−3.58%	15.17%	0.297 (ttS)
	**mOOcIP**	** 0.432 **	0.176	−0.784	−2.92%	18.85%	0.518 (tW)
**Distance from epidermis**	**mOF**	** 0.467 **	−0.050	−0.840	−7.85%	17.73%	** 0.003 (ttS) **
	**mP**	** 0.758 **	−0.061	−0.343	−2.91%	10.72%	0.505 (ttS)
	**mCS**	** 0.304 **	−0.163	0.125	4.63%	11.17%	** 0.019 (ttS) **
	**mOOcSO**	** 0.454 **	0.312	−0.503	−5.73%	17.62%	0.184 (ttS)
	**mOOcSP**	0.171	0.163	−0.757	−8.74%	19.37%	** 0.008 (ttS) **
	**mOOcIO**	** 0.359 **	−0.124	−0.647	−1.24%	16.74%	0.306 (tW)
	**mOOcIP**	** 0.276 **	−0.085	−0.470	−5.42%	16.85%	0.527 (ttS)

ttS—paired Student’s *t*-test; tW—Wilcoxon signed-rank test; mOF—frontal belly of the occipitofrontalis muscle; mP—procerus muscle; mCS—corrugator supercilii muscle; mOOcSO—orbicularis oculi muscle, superior eyelid orbital part; mOOcSP—orbicularis oculi muscle, superior eyelid palpebral part; mOOcIO—orbicularis oculi muscle, inferior eyelid orbital part; mOOcIP—orbicularis oculi muscle, inferior eyelid palpebral part.

**Table 5 toxins-17-00595-t005:** Comparison of the mean and standard deviation of muscle thickness and muscle-to-epidermis distance with values reported in the literature.

	Thickness [mm]		Distance from Epidermis [mm]	
	Study Results	Literature Review	Study Results	Literature Review
**mOF**	1.06 ± 0.52	F: 2.88 ± 0.68 [16] M: 2.35 ± 0.79 [16] RF: 2.88 ± 0.56 [17] RM: 2.20 ± 1.00 [17] LF: 2.86 ± 0.58 [17] LM: 2.27 ± 0.95 [17] 2.2 ± 0.5 [21]	3.12 ± 0.88	no data
**mP**	1.35 ± 0.33	0.56 [22] 1.1 ± 0.5 [23]	3.02 ± 0.66	3.8 ± 0.7 [23]
**mCS**	2.74 ± 0.67	R: 1.64 ± 0.39 [18] L: 1.6 ± 0.42 [18] 1.6 [19]	3.82 ± 0.66	no data
**mOOcSO**	1.14 ± 0.26	F: 0.84 ± 0.19 [16] M: 1.00 ± 0.25 [16] RF: 0.76 ± 0.26 [17] RM: 1.02 ± 0.16 [17] LF: 0.72 ± 0.22 [17] LM: 1.07 ± 0.17 [17]	1.88 ± 0.58	no data
**mOOcSP**	0.95 ± 0.25	no data	1.10 ± 0.03	no data
**mOOcIO**	1.62 ± 0.37	no data	1.69 ± 0.46	F: 2.5 ± 0.6 [24] M: 2.5 ± 0.5 [24]
**mOOcIP**	1.56 ± 0.41	no data	1.67 ± 0.41	no data

R—right side of the face; L—left side of the face; F—female; M—male; mOF—frontal belly of the occipitofrontalis muscle; mP—procerus muscle; mCS—corrugator supercilii muscle; mOOcSO—orbicularis oculi muscle, superior eyelid orbital part; mOOcSP—orbicularis oculi muscle, superior eyelid palpebral part; mOOcIO—orbicularis oculi muscle, inferior eyelid orbital part; mOOcIP—orbicularis oculi muscle, inferior eyelid palpebral part.

## Data Availability

The raw data supporting the conclusions of this article will be made available by the authors on request.

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
