# Peer review of "Ultrasound Evaluation of Upper Facial Muscles to Guide Botulinum Toxin Application"

_toxins, 2025, doi:10.3390/toxins17120595_

Round 1
Reviewer 1 Report
Comments and Suggestions for Authors
Thank you for sending me this interesting article, which evaluates the role of ultrasound in the upper facial muscles for preventing potential complications arising from botulinum toxin injections in the forehead. This manuscript represents a valuable contribution to the field of ultrasound-guided aesthetic medicine, focusing on the need for individualized planning of botulinum toxin injections. The study provides ultrasound data of the upper facial muscles and demonstrates significant anatomical variability according to sex, BMI, and laterality.
However, after carefully reading the manuscript, we have some comments and concerns that we will share with the authors for their consideration and response.
METHODOLOGY
How many participants were invited to participate in the study, and how many declined? All 127 patients were enrolled and completed the study, but it is not detailed whether there were any participant withdrawals. Line 293 mentions frequent withdrawals, but these are not quantified. Could there have been a recruitment risk, given that the patients were described as volunteers (lines 66, 338)? Where were the participants selected from? Were they healthy volunteers or patients who came to the university for another reason?
The study includes 127 participants, but no justification is provided for the sample size. Was a pre-calculation of the sample size performed? The study included 103 women and 24 men, resulting in a critical gender imbalance. While the authors acknowledge this limitation, they do not quantify its impact.
Claustrophobia is mentioned as an exclusion criterion (line 334), but this criterion for undergoing an ultrasound is unclear, given that it is an open procedure.
Measurements were repeated three times for each muscle (line 349). However, no information is provided on how the observers performed these measurements. It is unknown whether there were a single operator or multiple operators. Was an interclass correlation coefficient (ICC) study performed to assess intraobserver reliability?
The methodology lacks information on the observers' knowledge and training, as well as their years of experience and dedication to musculoskeletal ultrasound. The learning curve for performing the ultrasound is not addressed. The lack of professional training is mentioned as a barrier (lines 53-56), but no further details are provided regarding the necessary training requirements for the observers, considering that ultrasound is highly dependent on the practitioner.
On line 294, it is noted that the technique often caused discomfort to participants due to the application of gel near the eyes, but the number of participants with this complication is not quantified. The figures are very illustrative of the measurement methods. However, a more detailed description of the measurement protocol is needed: the position and angle of the participant's head, the muscle status (relaxed or contracted), the compression exerted on the tissues (line 338), the identification of anatomical landmarks (lines 72-73), etc.
RESULTS
The results section should provide a more in-depth clinical interpretation of the statistically significant findings and clarify when these differences are clinically relevant (minimum clinically important difference). An effect size calculation is not presented.
This study yielded multiple statistically significant findings. However, the minimum clinically important difference for each measurement is not defined. The clinical relevance of the findings is unknown; that is, is it clinically significant for botulinum toxin injection planning if the upper facial muscles are tenths of a millimetre thicker or deeper? Most of the significant results for thickness or depth are less than 0.1 mm. What is the clinical significance of this?
In the statistical analysis, were multivariable regression models explored to assess the pooled predictive value of sex, age, BMI, etc.? In the Abstract (lines 18-19), it is stated that the asymmetry coefficients reached 40%, but it should be clarified that this is a quartile deviation, not a deviation from the mean or median.
DISCUSSION
It appears that the study was conducted in a Central European country. The participants were likely of homogeneous Caucasian ancestry. It would be advisable to describe the limitations of generalizing the study to other (multi) ethnic groups.
The study is limited to participants aged 21–40, but in practice, botulinum toxin injections for facial rejuvenation are used across a much wider age range. The expected results in an older population, considering, for example, age-related muscle changes, are not discussed.
A specific discussion is needed to determine the clinical implications of the injection technique based on ultrasound data. In this regard, more detailed recommendations on injection depth and location, based on ultrasound findings, should be provided, along with a mapping of safe injection zones and, ultimately, a more precise, useful, and practical protocol for performing ultrasound-guided injections. The reader misses a clinical algorithm that integrates ultrasound findings for the planning and execution of injections in a clinical setting.
There is a lack of argumentation and debate regarding the potential clinical outcomes compared between those obtained using ultrasound guidance versus traditional results based on anatomical landmarks.
In lines 308-310 and 315-317, it is argued that muscle ultrasound allows for more effective procedure planning and can help identify botulinum toxin injection points, as well as monitor the procedure intraoperatively to minimize the risk of complications. One of the study's objectives was to provide data for the standardization of ultrasound examination and individualized injection techniques (lines 61-64). These are all ambitious objectives that are not specifically addressed by the authors' study, as their findings do not clearly demonstrate how they translate to clinical practice.
Author Response
METHODOLOGY
Comment #1: How many participants were invited to participate in the study, and how many declined? All 127 patients were enrolled and completed the study, but it is not detailed whether there were any participant withdrawals. Line 293 mentions frequent withdrawals, but these are not quantified. Could there have been a recruitment risk, given that the patients were described as volunteers (lines 66, 338)? Where were the participants selected from? Were they healthy volunteers or patients who came to the university for another reason?
Response #1: Thank you for this comment and for drawing attention to the recruitment process. All participants in our study were healthy volunteers recruited specifically for this project, and not patients presenting for any clinical reason. Recruitment was carried out through email notifications, social media groups, and other accessible community channels. Therefore, all individuals enrolled in the study came from the general population of Warsaw (to prevent selection bias). Regarding the numerical information on incomplete measurements and sample sizes, these details were summarized in Table 1. Variation in the number of observations between muscles reflects incomplete measurement sets for some individuals during the examination session. The reduced observation counts in Table 1 resulted from one type of within-session discontinuation, where only part of the planned examination was completed. These partial but analyzable measurements were included in the study and account for the variation in sample sizes across muscles. A second type of interruption also occurred, involving volunteers who discontinued the examination almost immediately when the gel pad approached the peri-ocular region. Because these cases yielded only fragmentary, non-measurable images, they did not contribute any data to the analysis and therefore do not appear in Table 1. This very early discontinuation was observed exclusively among male volunteers; had these individuals completed the examination, the number of men with usable data would have exceeded forty. This second form of early withdrawal, which contributed to the disproportionately lower number of male participants, has already been explained in the lines of the Discussion section referenced by the Reviewer.
The Methods and Results sections have been revised to explicitly report these details.
Comment #2: The study includes 127 participants, but no justification is provided for the sample size. Was a pre-calculation of the sample size performed? The study included 103 women and 24 men, resulting in a critical gender imbalance. While the authors acknowledge this limitation, they do not quantify its impact.
Response #2: Thank you for pointing this out.
Sample size justification: No formal a priori sample size calculation was performed. The study was designed as an exploratory and descriptive aimed at establishing reference ultrasound characteristics in a healthy population. Therefore, recruitment continued until a sufficiently large volunteer cohort was obtained to allow stable estimation of mean values and variability across measurements.
Post-hoc statistical power: Although no pre-study power calculation was conducted, we performed a post-hoc power estimation based on the observed sex differences in muscle thickness. For muscles where statistically significant sex-related differences were found (e.g., mP, mOF, mOOcSO, mOOcSP), the effect size was in the medium range (r ≈ 0.29). Given the group sizes (103 women and 24 men), the corresponding post-hoc power for detecting such effects exceeds 80%, indicating that the study was adequately powered to detect medium-to-large sex differences. In contrast, the study was not powered to detect small differences between sexes, and non-significant results should therefore be interpreted with caution. The uneven sex distribution resulted from volunteer-based recruitment and while this remains a limitation, its impact is primarily on the ability to detect subtle sex differences, and not on the descriptive aim of the study.
We have clarified these points in the Methods (sample size justification) and Discussion (post-hoc statistical power) sections.
Comment #3: Claustrophobia is mentioned as an exclusion criterion (line 334), but this criterion for undergoing an ultrasound is unclear, given that it is an open procedure.
Response #3: Thank you for this comment. We agree that claustrophobia is not an exclusion criterion for ultrasound in general. In our study, however, the imaging protocol required participants to keep their eyes closed for an extended period while a gel pad was placed directly over the eyes. This setup can induce discomfort or claustrophobic sensations in some individuals. For this reason, claustrophobia was included as a precautionary exclusion criterion.
Comment #4: Measurements were repeated three times for each muscle (line 349). However, no information is provided on how the observers performed these measurements. It is unknown whether there were a single operator or multiple operators. Was an interclass correlation coefficient (ICC) study performed to assess intraobserver reliability?
Response #4: Thank you for raising this important point regarding measurement reliability. All ultrasound measurements were performed by a single experienced operator. After data collection, all images and measurements were independently reviewed by a second senior expert to ensure consistency and adherence to the protocol. No discrepancies requiring re-evaluation were identified. Because all primary measurements were acquired by a single operator, and the repeated measurements served to increase precision rather than to quantify rater variability, a formal intraobserver ICC analysis was not performed. The purpose of performing multiple measurements has now been explicitly incorporated into the Materials and Methods section, along with a detailed description of the measurement and review procedure.
Comment #5: The methodology lacks information on the observers' knowledge and training, as well as their years of experience and dedication to musculoskeletal ultrasound. The learning curve for performing the ultrasound is not addressed. The lack of professional training is mentioned as a barrier (lines 53-56), but no further details are provided regarding the necessary training requirements for the observers, considering that ultrasound is highly dependent on the practitioner.
Response #5: Thank you for this comment regarding observer training and expertise. All ultrasound measurements were performed by an operator with 5 years of experience in facial ultrasound and 10 years of experience in diagnostic ultrasound overall. The operator’s extensive training and practice in musculoskeletal ultrasound ensured reliable image acquisition. The collected data were reviewed by a second expert, who had 30 years of experience in facial ultrasound and 35 years of experience in ultrasound in general. Although formal documentation of the learning curve was not included, the operator’s expertise and repeated measurements ensured high measurement precision. The Materials and Methods section has been updated to provide more explicit information on operators’ qualifications, training, and experience, as well as on the review process, to address this point.
Comment #6: On line 294, it is noted that the technique often caused discomfort to participants due to the application of gel near the eyes, but the number of participants with this complication is not quantified. The figures are very illustrative of the measurement methods. However, a more detailed description of the measurement protocol is needed: the position and angle of the participant's head, the muscle status (relaxed or contracted), the compression exerted on the tissues (line 338), the identification of anatomical landmarks (lines 72-73), etc.
Response #6: Thank you for this suggestion. As noted in our response to Reviewer Comment #1, the exact numbers of incomplete measurements are provided in Table 1, which includes all interrupted examinations—both those where the reason was reported by the participant and those without. Regarding the Reviewer’s request for a more detailed description of the measurement protocol, all relevant information has now been added to the manuscript. The figures are also accompanied by a more detailed description of the measurement protocol.
RESULTS
Comment #7: The results section should provide a more in-depth clinical interpretation of the statistically significant findings and clarify when these differences are clinically relevant (minimum clinically important difference). An effect size calculation is not presented.
Response #7: Thank you for this comment. Effect sizes are now consistently reported across all relevant analyses. For the associations with age and BMI, the previously reported correlation coefficients serve directly as effect size measures. For sex-related comparisons, an additional column reporting effect sizes has been added to Table 3. For side-to-side comparisons, where an additional column could not be accommodated, effect sizes for statistically significant results have been incorporated directly into the Results text. The Methods section has also been updated to explicitly describe the effect size metrics used. Further discussion of clinical relevance and minimum clinically important differences is provided in the response to Comment #8.
Comment #8: This study yielded multiple statistically significant findings. However, the minimum clinically important difference for each measurement is not defined. The clinical relevance of the findings is unknown; that is, is it clinically significant for botulinum toxin injection planning if the upper facial muscles are tenths of a millimetre thicker or deeper? Most of the significant results for thickness or depth are less than 0.1 mm. What is the clinical significance of this?
Response #8: We agree that the interpretation of statistically significant findings requires understanding the minimum clinically important difference (MCID). However, an MCID for ultrasound-based thickness or depth of upper facial muscles has not been defined in the literature, either for between-subject differences or within-subject asymmetries. Because of no MCID, it was not possible to predefine a clinically meaningful threshold for our measurements. Consequently, the study was intentionally designed as an exploratory descriptive project, aimed at establishing normative data and quantifying natural variability rather than testing hypotheses related to clinical significance. The absence of an established MCID also meant that a priori sample size calculation and power analysis could not be performed, as these require a predefined effect size. We hope that the results of the present study will provide a foundation for defining MCID values in future research. Once such thresholds are established, they will allow clinically oriented hypothesis testing and more targeted sample size estimation.
In addition, we agree that the clinical interpretation of our anatomical findings requires separate consideration. Because clinical interpretation extends beyond the scope of the Results section, which is limited to presenting the measurements themselves, we expanded the Discussion to include a dedicated paragraph outlining preliminary clinical implications of the observed muscle dimensions and asymmetries. This section also clarifies how the present dataset may support future studies designed to define clinically meaningful thresholds, including the MCID.
Comment #9: In the statistical analysis, were multivariable regression models explored to assess the pooled predictive value of sex, age, BMI, etc.? In the Abstract (lines 18-19), it is stated that the asymmetry coefficients reached 40%, but it should be clarified that this is a quartile deviation, not a deviation from the mean or median.
Response #9: Thank you for this helpful comment. We did explore multivariable regression models during the analysis; however, only 3 out of 14 models met the statistical assumptions. Moreover, the models that could be reliably fitted did not provide additional insights beyond those already presented. For this reason, we initially chose not to include them in the manuscript.
The 40% value in the Abstract represents the relative side-to-side asymmetry coefficient, not a quartile deviation or deviation from the mean/median. To eliminate any ambiguity, we have revised the Abstract.
DISCUSSION
Comment #10: It appears that the study was conducted in a Central European country. The participants were likely of homogeneous Caucasian ancestry. It would be advisable to describe the limitations of generalizing the study to other (multi) ethnic groups.
Response #10: Thank you for this comment. We acknowledge that the study population was relatively ethnically homogeneous, and we have now clarified in the Limitations section that this may limit generalizability. Future studies including more diverse populations would help capture broader anatomical variability.
Comment #11: The study is limited to participants aged 21–40, but in practice, botulinum toxin injections for facial rejuvenation are used across a much wider age range. The expected results in an older population, considering, for example, age-related muscle changes, are not discussed.
Response #11: We thank the Reviewer for this important observation. The age range of 21–40 years was deliberately selected to establish normative reference values in young, healthy adults with minimal age-related degenerative changes, thereby providing a clear baseline for future comparisons with older cohorts. Nonetheless, we have added a dedicated paragraph in the Discussion to contextualize our findings with age-related muscle and skin changes, highlighting their potential implications for clinical practice in older patients.
Comment #12: A specific discussion is needed to determine the clinical implications of the injection technique based on ultrasound data. In this regard, more detailed recommendations on injection depth and location, based on ultrasound findings, should be provided, along with a mapping of safe injection zones and, ultimately, a more precise, useful, and practical protocol for performing ultrasound-guided injections. The reader misses a clinical algorithm that integrates ultrasound findings for the planning and execution of injections in a clinical setting.
Response #12: We appreciate the reviewer’s suggestion. While the idea of developing a universal ultrasound-guided injection protocol is understandable, our results clearly demonstrate pronounced anatomical variability between patients, and even between sides of the same face. Because of this variability, fixed “safe zones” or standardized injection maps may be misleading or inaccurate. Therefore, instead of proposing a single protocol, our study highlights that safety in aesthetic toxin injections is best achieved through individualized, real-time ultrasound assessment, which allows personalized selection of injection points rather than relying on predefined schemes.
To address this point, we have expanded the Discussion section to include clinical implications derived from previous studies, such as the observation that men typically require higher toxin doses due to greater muscle mass. We have also referred to cadaver-based research demonstrating that ultrasound-guided injections improve placement accuracy compared with landmark-only techniques (which also responds to the Reviewer’s comment #13).
Comment #13: There is a lack of argumentation and debate regarding the potential clinical outcomes compared between those obtained using ultrasound guidance versus traditional results based on anatomical landmarks.
Response #13: We thank the Reviewer for this comment. We have added a discussion paragraph highlighting that ultrasound guidance improves anatomical precision, enhances safety, and may optimize outcomes compared with landmark-based injections.
Comment #14: In lines 308-310 and 315-317, it is argued that muscle ultrasound allows for more effective procedure planning and can help identify botulinum toxin injection points, as well as monitor the procedure intraoperatively to minimize the risk of complications. One of the study's objectives was to provide data for the standardization of ultrasound examination and individualized injection techniques (lines 61-64). These are all ambitious objectives that are not specifically addressed by the authors' study, as their findings do not clearly demonstrate how they translate to clinical practice.
Response #14: We thank the Reviewer for this comment and fully agree that the objective stated in lines 61–64 must be clearly linked to the presented results. The exact wording of the objective (Introduction, lines 61–64) is:
„Additionally, this study aims to provide unified and reproducible data on the morphological parameters of the upper facial muscles, which can provide a basis for further standardization of US examinations and individualized injection techniques.”
This objective has been completely fulfilled as follows:
- Reproducible US examination is achieved by transducer positions illustrated in the Methods and Materials section and by entire upper facial musculature ultrasound images in the Results section.
- Morphological data are delivered with exact descriptive statistics for thickness and depth of all examined muscles.
Then, there is the deliberately chosen phrasing „a basis for further (…)” indicates that the present normative study supplies the essential reference dataset and standardized imaging technique, not a finalized clinical algorithm (which would require additional prospective trials).
Regarding mentioned lines from the Discussion section, these sentences do not represent study objectives but constitute general concluding statements. We assume that the Reviewer is highlighting the issue of applying our ultrasound data in a clinical context. These aspects have already been thoroughly addressed in our responses to the Reviewer’s previous comments, where we added dedicated paragraphs and references. We hope that the above clarifications, together with the revisions and additions made in response to the Reviewer’s earlier comments, convincingly demonstrate that the ultrasound findings presented in this study accurately reflect anatomical variability and support their potential translation into clinical practice.
Reviewer 2 Report
Comments and Suggestions for Authors
Aim of this very interesting paper was to familiarize oneself with the possibilities of ultrasonographic evaluation of the upper facial muscles, focusing on interindividual and intraindividual anatomical variations to reduce the risk of complications after botulinum toxin treatment.
Enrolling 127 participants (103 women and 24 men), the study provides normative ultrasound parameters for the upper facial muscles in healthy adults and demonstrates significant anatomical variability depending on sex, BMI, and facial laterality, supporting individualized ultrasound-guided approaches for botulinum toxin injection.
The results, conclusions, and study protocol were presented precisely and with good scientific resonance, providing numerous scientific and clinical paramenters for the treatment of the upper facial muscles with botulinum toxin. No improvement regarding methodology but, however, before publication, manuscript requires some modifications:
1) Improve the English and
2) Improve the bibliography by including major number of recent papers (only 5/32 are less than 5 years old).
Conclusions were consistent with the evidence and arguments presented , providing numerous scientific and clinical paramenters for the treatment of the upper facial muscles with botulinum toxin. However, in this section, sentences should contain the "may" or "could" without making clear scientific and clinical sentences, considering the appropriate limitations of the study indicated by the authors (for example, line 200 "This study may/could established normative ultrasound values...", line 201 "...results may/could show that muscle.....").
Comments on the Quality of English LanguageImprove the English.
Author Response
Reviewer's comments:
1) Improve the English and
2) Improve the bibliography by including major number of recent papers (only 5/32 are less than 5 years old).
Conclusions were consistent with the evidence and arguments presented , providing numerous scientific and clinical paramenters for the treatment of the upper facial muscles with botulinum toxin. However, in this section, sentences should contain the "may" or "could" without making clear scientific and clinical sentences, considering the appropriate limitations of the study indicated by the authors (for example, line 200 "This study may/could established normative ultrasound values...", line 201 "...results may/could show that muscle.....").
Response:
We thank the reviewer for these comments. (1) The entire manuscript has undergone a thorough English-language revision to improve clarity and scientific style. (2) The bibliography has been substantially updated and expanded to include a larger proportion of recent publications, especially from the last five years. Several contemporary studies on facial anatomy, ultrasound assessment, and botulinum toxin treatment have now been incorporated into the Introduction and Discussion.
Regarding the Conclusions section, we agree that the wording should reflect the exploratory and descriptive nature of the study. Accordingly, the statements that could be interpreted as definitive have been revised to include more cautious phrasing (e.g., “may” or “could”), in line with the reviewer’s recommendations.
Reviewer 3 Report
Comments and Suggestions for Authors
The article addresses the applicability of ultrasound to evaluate facial muscles. The technique could improve botulinum toxin procedures.
TITLE
Ultrasound Evaluation of Upper Facial Muscles to Prevent Botulinum Toxin Complications
-Authors should reflect whether botulinum toxin complications should be in the title, considering that these complications and procedures have not been tested in this work. The work performed a series of measurements of muscles, proving that ultrasound is efficacious in this analysis. Since these muscles are targets of treatment with botulinum toxin, it is possible to infer that ultrasound would prevent complications of botulinum toxin. But this cannot extrapolate an inference, since the ability to prevent complications was not tested. Ultrasound Evaluation of Upper Facial Muscles to Guide Botulinum Toxin Application (?)
ABSTRACT
Lack of age correlation confirmed age group consistency.
-Correlation between "age" and what?
FIGURES
If possible, please provide separate figures for each muscle, each accompanied by a proper legend, rather than combining two different muscles under the same legend.
Line 377-378: Dependent sample comparisons were performed using the Student’s t-test
-Paired-samples Student t test
TABLES
I suggest avoiding the excessive use of bold and capital letters in the tables
DISCUSSION
-I think Table 5 should not be in the discussion section. Would it not be better if the authors presented brief sentences summarizing the comparison of their results with the literature reports instead of inserting table 5 in the discussion? Maybe table 5 should be moved to supplementary material section.
-Lines 250-251: The measurements of the orbital part thickness (mOOcSO) and depth (mOOcIO) align with values reported in the literature (Table 5), confirming that ultrasound enables…
-Instead of citing Table 5 in parentheses in the sentence above, the references supporting this comparison could be cited.
Author Response
TITLE
Comment #1: Ultrasound Evaluation of Upper Facial Muscles to Prevent Botulinum Toxin Complications
-Authors should reflect whether botulinum toxin complications should be in the title, considering that these complications and procedures have not been tested in this work. The work performed a series of measurements of muscles, proving that ultrasound is efficacious in this analysis. Since these muscles are targets of treatment with botulinum toxin, it is possible to infer that ultrasound would prevent complications of botulinum toxin. But this cannot extrapolate an inference, since the ability to prevent complications was not tested. Ultrasound Evaluation of Upper Facial Muscles to Guide Botulinum Toxin Application (?)
Response #1: Thank you for this insightful comment. We agree that the previous title could imply that botulinum toxin–related complications were directly evaluated in this study. As suggested by the reviewer, we have updated the title to: “Ultrasound Evaluation of Upper Facial Muscles to Guide Botulinum Toxin Application.” We appreciate this helpful recommendation.
ABSTRACT
Comment #2: Lack of age correlation confirmed age group consistency.
-Correlation between "age" and what?
Response #2: Thank you for pointing out this ambiguity. We clarified the sentence in the Abstract. Age was analyzed for correlation with all ultrasound measurements (muscle thickness and depth), and no significant associations were found. The sentence has been revised (“Age showed no significant correlation with muscle thickness or depth, supporting the internal consistency of the studied age group.”)
FIGURES
Comment #3: If possible, please provide separate figures for each muscle, each accompanied by a proper legend, rather than combining two different muscles under the same legend.
Response #3: Thank you for this helpful suggestion. We have separated the figures so that each muscle is now presented individually with its own dedicated legend. This modification improves clarity and readability, and we appreciate the reviewer’s comment.
Comment #4: Line 377-378: Dependent sample comparisons were performed using the Student’s t-test
-Paired-samples Student t test
Response #4: Thank you for the correction. We have updated the Methods section accordingly.
TABLES
Comment #5: I suggest avoiding the excessive use of bold and capital letters in the tables
Response #5: Thank you for this suggestion. The formatting of all tables has been revised to remove excessive use of bold and capital letters. Only the BMI category labels remain in bold and capital letters for clarity, while all other table headings have been converted to lowercase.
DISCUSSION
Comment #6: -I think Table 5 should not be in the discussion section. Would it not be better if the authors presented brief sentences summarizing the comparison of their results with the literature reports instead of inserting table 5 in the discussion? Maybe table 5 should be moved to supplementary material section.
Response #6: Thank you for this comment. Following the reviewer’s suggestion, we attempted to replace Table 5 with a narrative summary. However, due to the number of studies and the variability in the reported ultrasound values, the text-only version became long, difficult to follow, and less informative than the tabular format. We therefore propose to retain Table 5 in the main Discussion, as it allows readers to efficiently compare our results with previously published data.
Comment #7: -Lines 250-251: The measurements of the orbital part thickness (mOOcSO) and depth (mOOcIO) align with values reported in the literature (Table 5), confirming that ultrasound enables…
-Instead of citing Table 5 in parentheses in the sentence above, the references supporting this comparison could be cited.
Response #7: Thank you for this suggestion. The sentence has been revised accordingly: instead of referring to Table 5, we now cite the specific literature sources that support the comparison.
Reviewer 4 Report
Comments and Suggestions for Authors
Toxins 3995331
This is an interesting and well-written manuscript. I have several concerns, which are described below.
Introduction
Line 32 The date is incorrect. The first approval of Botox was in December 1989 for strabismus and blepharospasm. Approval for the treatment of glabellar lines was in 2002 as Botox Cosmetic. Approval for treatment of forehead wrinkles can be found from the FDA web site https://www.accessdata.fda.gov/scripts/cder/daf/index.cfm
Citations [1] and [2] are almost 10 years old. More recent citations should be used.
Lines 37 & 40 Citations are needed for these numbers
Line 42 Is citation [4] correct here?
Lines 55-57 Citations needed
Materials and Methods
Was this study registered on a clinical trial database? This should be the case.
Results
For Figure 2, the legend does not mention what panels I to P are showing, apart from the coloured arrow heads.
For Figures 3 & 4 information about the relative colours of the data shown needs to be given. What do the different colours mean? Or are these different colours intended just to show the different muscles?
Discussion
Lines 213-216 Citation needed
In their discussion of data from other studies, the authors have not indicated the ages of patients studied. This should be included since age has a significant influence on muscles of the face.
The authors have not discussed the practicality of using ultrasound guidance for aesthetic toxin treatments. One of the major advantages seen by many patients is the speed with which toxin treatments may be given. Using ultrasound significantly increases this time. Also, ultrasound requires significant training of the injector: I doubt whether the majority of aesthetic injectors in the world have such a training. The authors should add discussion on these two aspects.
The authors have not included or discussed other literature relating to the muscle anatomy of the face and BoNT injections. For example:
Raveendran, S. S., & Anthony, D. J. (2020). Classification and Morphological Variation of the Frontalis Muscle and Implications on the Clinical Practice. Aesthetic Plast Surg, 45(1), 164–170. https://doi.org/10.1007/s00266-020-01937-2
The authors should comment on other studies such as this.
Author Response
Introduction
Comment #1: Line 32 The date is incorrect. The first approval of Botox was in December 1989 for strabismus and blepharospasm. Approval for the treatment of glabellar lines was in 2002 as Botox Cosmetic. Approval for treatment of forehead wrinkles can be found from the FDA web site https://www.accessdata.fda.gov/scripts/cder/daf/index.cfm
Response #1: We are grateful for the reviewer’s remark. The date has been corrected.
Comment #2: Citations [1] and [2] are almost 10 years old. More recent citations should be used.
Response #2: Thank you for this suggestion. The cited references [1] and [2] have been updated with more recent publications.
Comment #3: Lines 37 & 40 Citations are needed for these numbers
Response #3: Thank you for pointing this out. We have added missing citations.
Comment #4: Line 42 Is citation [4] correct here?
Response #4: We appreciate the reviewer’s helpful comment. We have changed this citation.
Comment #5: Lines 55-57 Citations needed
Response #5: Thank you for this valuable remark. We have now added the appropriate citations to lines 55–57. These references were previously included only in the Discussion, and we agree that adding them here improves clarity.
Materials and Methods
Comment #6: Was this study registered on a clinical trial database? This should be the case.
Response #6: We appreciate the reviewer’s comment. This study involves non-interventional, observational measurements of facial muscles using ultrasound and does not constitute a clinical trial, and was therefore not registered in a clinical trial database.
Results
Comment #7: For Figure 2, the legend does not mention what panels I to P are showing, apart from the coloured arrow heads.
Response #7: Thank you for this comment. The figure has been revised: each muscle is now shown separately, and the figure legend has been reorganized to clearly describe the content of panels.
Comment #8: For Figures 3 & 4 information about the relative colours of the data shown needs to be given. What do the different colours mean? Or are these different colours intended just to show the different muscles?
Response #8: We thank the reviewer for this comment. The different colors in Figures 3 and 4 are used solely for visual distinction of the muscles and do not represent any quantitative or categorical differences.
Discussion
Comment #9: Lines 213-216 Citation needed
Response #9: Thank you for pointing this out. A citation has been added to support the statements in lines 213–216.
Comment #10: In their discussion of data from other studies, the authors have not indicated the ages of patients studied. This should be included since age has a significant influence on muscles of the face.
Response #10: We thank the reviewer for this comment. We agree that patient age is an important factor influencing facial muscles. A dedicated paragraph has been added to the Discussion section.
Comment #11: The authors have not discussed the practicality of using ultrasound guidance for aesthetic toxin treatments. One of the major advantages seen by many patients is the speed with which toxin treatments may be given. Using ultrasound significantly increases this time. Also, ultrasound requires significant training of the injector: I doubt whether the majority of aesthetic injectors in the world have such a training. The authors should add discussion on these two aspects.
Response #11: Thank you very much for this insightful comment. We agree that the practicality of ultrasound guidance in aesthetic toxin procedures requires clearer discussion. In the revised manuscript, we have expanded the Discussion to address both aspects related to procedure duration and operator training.
Because it was not entirely clear whether the reviewer referred to the time of the injection itself or to the overall duration of the visit, we would like to clarify both points. In aesthetic practice, the injection itself does not need to take longer when using ultrasound, as the toxin can be administered either under real-time guidance or after a pre-procedural ultrasound mapping of the injection sites. However, the overall procedure time is increased due to the additional steps required for ultrasound imaging and site localization. Incorporating ultrasound into the appointment does extend the total visit time, typically by approximately 10–15 minutes. We note that this added time may be offset by greater accuracy in targeting the muscle belly and selecting appropriate dosing.
We also expanded the discussion regarding operator training. As the reviewer correctly notes, ultrasound-guided injections require appropriate expertise, and not all aesthetic injectors currently possess these skills. We therefore emphasize in the revised text that successful use of facial ultrasound depends on dedicated training in both ultrasound anatomy and image acquisition.
Comment #12: The authors have not included or discussed other literature relating to the muscle anatomy of the face and BoNT injections. For example:
Raveendran, S. S., & Anthony, D. J. (2020). Classification and Morphological Variation of the Frontalis Muscle and Implications on the Clinical Practice. Aesthetic Plast Surg, 45(1), 164–170. https://doi.org/10.1007/s00266-020-01937-2
The authors should comment on other studies such as this.
Response #12: Thank you for this valuable comment. We agree that relevant anatomical literature should be more thoroughly incorporated into the Discussion. In the revised manuscript, we added a dedicated paragraph summarizing key cadaveric evidence on the anatomical variability of upper-face muscles relevant to botulinum toxin procedures.
Round 2
Reviewer 1 Report
Comments and Suggestions for Authors
We thank the authors for their extensive and detailed report in response to our comments and suggestions. The new insertions they have added to their manuscript undoubtedly improve the quality and relevance of their research.
The authors have responded extensively to our comments from the first round, clarifying the recruitment process and adding methodological details. In the revised version, the authors have improved transparency regarding recruitment and the study's limitations. The manuscript now presents more clearly the normative ultrasound values ​​for facial musculature in young Caucasian adults. We also appreciate the new insertions regarding the observer who performed the measurements, their reliability and experience, as well as the research protocol (lines 426-460). From a statistical viewpoint, effect sizes have been included.
However, the revised manuscript confirms that this is a purely descriptive and anatomical study with some limitations that are difficult to overcome given the methodological nature of the research.
Comment #8. The authors explicitly admit that there is no defined "Minimum Clinically Important Difference" (MCID) and that their study cannot establish one. They report very small statistical differences, on the order of tenths of a millimetre (e.g., 0.1–0.2 mm in muscle thickness between sexes), but this study does not justify how this can change the clinical practice of botulinum toxin injection for aesthetic procedures in the upper face.
Comment #1 and Comment #11. The fact that the participants were healthy volunteers recruited through social media introduces a self-selection bias. Furthermore, the 21–40 age range precisely excludes the actual target population for toxin treatments (patients aged 40–60 with atrophic muscle changes). This drastically limits the usefulness of these "normative values" for real-world clinical practice. How is the loss of data in men due to discomfort justified (Response #1)? This suggests selection bias.
Comment #2. It is acknowledged that a sample size calculation was not performed a priori and that the gender imbalance (103 women versus 24 men) constitutes a significant limitation resulting from convenience sampling.
Comments #12, #13, and #14. The new title of the manuscript states ("...to guide the application of botulinum toxin"), but the study does not offer practical guidance or an algorithm to guide botulinum toxin injections. It simply notes that ultrasound can improve preoperative or intraoperative accuracy and that "variability exists," something already known, but it does not define how this translates into actual clinical practice. If the main conclusion is that "there is great variability, so each patient should be individually assessed by ultrasound," the normative values ​​presented lose, to some extent, their true practical value. There is no clinical algorithm to improve its use, regardless of the results of this exploratory and descriptive study on normal sonographic anatomy in a specific group of young adults, predominantly women and Caucasian.
Lines 294 to 358. The newly added section on clinical implications is very interesting but essentially remains speculative (based on previous literature on dosage in men) and is not directly derived from the data measured in this study.
Author Response
General clarification
Before addressing the comments, we would like to emphasize that our work is not a clinical study and did not involve any botulinum toxin injections. Consequently, there is no methodological basis for proposing injection protocols, dosing recommendations, or clinical algorithms. The aim of this study was strictly anatomical: to evaluate the sonographic appearance and measurable variability of the upper-face muscles that are commonly targeted in aesthetic practice. In this context, the study provides a standardized imaging and measurement protocol, but not a protocol for injections. Establishing clinical recommendations would require outcome-based trials with actual injections and randomized comparison of techniques, which lies outside the scope of this exploratory anatomical investigation.
Comment #1: Comment #8. The authors explicitly admit that there is no defined "Minimum Clinically Important Difference" (MCID) and that their study cannot establish one. They report very small statistical differences, on the order of tenths of a millimetre (e.g., 0.1–0.2 mm in muscle thickness between sexes), but this study does not justify how this can change the clinical practice of botulinum toxin injection for aesthetic procedures in the upper face.
Response #1: We agree that small point-wise differences in muscle thickness (e.g., 0.1–0.2 mm) cannot be directly translated into changes in clinical practice. Importantly, our measurements represent single cross-sectional values of muscle thickness and depth at a predefined point, whereas clinically relevant differences relate primarily to overall muscle volume, architecture, and the three-dimensional course of the muscle fibers, rather than to isolated linear measurements. Thus, our intention was not to define clinically actionable thresholds; rather, we sought to document anatomical variability and confirm the utility of high-resolution ultrasound in capturing it. A comprehensive understanding of clinically relevant thresholds would require volumetric and geometric characterization of individual muscles (their overall volume, fiber orientation, morphology and depth variability along their entire course). Only multidimensional anatomical parameterization can be meaningfully correlated with treatment outcomes and serve as a basis for defining true MCID values or clinically applicable injection guidelines. This point has been added to the Discussion as an example of future work.
Comment #2: Comment #1 and Comment #11. The fact that the participants were healthy volunteers recruited through social media introduces a self-selection bias. Furthermore, the 21–40 age range precisely excludes the actual target population for toxin treatments (patients aged 40–60 with atrophic muscle changes). This drastically limits the usefulness of these "normative values" for real-world clinical practice. How is the loss of data in men due to discomfort justified (Response #1)? This suggests selection bias.
Response #2: We agree that recruiting volunteers through social media introduces an element of self-selection bias; however, in the context of establishing normative anatomical reference values, younger healthy adults are the standard population used in such studies. Norms are conventionally derived from structurally intact tissues, because age-related atrophy introduces substantial variability that is not intrinsic to the anatomy itself. Including participants aged 40–60 would in fact increase, rather than reduce, selection bias, as this population is more heterogeneous with respect to prior cosmetic procedures, lifestyle-related tissue changes, hormonal status, and chronic diseases. In contrast, in young adults, who have not yet had the opportunity to counteract age-related changes, these confounding factors are minimal, allowing a more reliable characterization of baseline anatomical variability.
Regarding the loss of data in men, we acknowledged this as a limitation. However, this limitation does not undermine the representativeness of the entire sample in the context of aesthetic practice. In our study, men constitute 18.9% of the final cohort, which aligns closely with international data showing that botulinum toxin procedures in 2024 were performed in men at a rate of 16% (ISAPS Global Survey 2024, p. 61, https://www.isaps.org/media/razfvmsk/isaps-global-survey-2024.pdf). Thus, the sex distribution in our study reflects real-world proportions of patients undergoing upper-face toxin treatments. The reduced number of male datasets affects primarily the statistical power for sex-specific comparisons, but not the generalizability of the overall normative values.
In summary, normative values are, by definition, established in healthy populations, and in this context age-related changes are treated analogously to pathological alterations that must be excluded to avoid confounding. Including older participants would only increase biological variability and amplify self-selection bias, as individuals in this age group may have already undertaken some actions to counteract facial ageing. Likewise, the observed sex-related selection bias arising from the lower availability of male volunteers reflects real-world patterns rather than methodological artefact. For these reasons, the current cohort provides an appropriate and clinically relevant basis for defining baseline anatomical variability.
Comment #3: Comment #2. It is acknowledged that a sample size calculation was not performed a priori and that the gender imbalance (103 women versus 24 men) constitutes a significant limitation resulting from convenience sampling.
Response #3: Both the absence of an a priori sample size calculation and the unequal sex distribution were clearly acknowledged in the manuscript. However, we would like to clarify the Reviewer’s use of the term “convenience sampling.” It is not entirely clear whether “convenience” refers to the convenience of the investigators or the participants. In either case, the imbalance did not arise from preferential enrollment but from volunteer availability. The proportion of men in our cohort closely matches recent global data (see Response #2), indicating that the sex distribution is in fact clinically representative rather than convenience-driven.
Methodologically, the total sample size in our study exceeds that of many previous ultrasonographic investigations of facial musculature, which often included 40 (or less) participants (see citations 21,22,23). Therefore, although we transparently acknowledge these constraints, they do not compromise the validity of our descriptive anatomical aims.
Comment #4: Comments #12, #13, and #14. The new title of the manuscript states ("...to guide the application of botulinum toxin"), but the study does not offer practical guidance or an algorithm to guide botulinum toxin injections. It simply notes that ultrasound can improve preoperative or intraoperative accuracy and that "variability exists," something already known, but it does not define how this translates into actual clinical practice. If the main conclusion is that "there is great variability, so each patient should be individually assessed by ultrasound," the normative values ​​presented lose, to some extent, their true practical value. There is no clinical algorithm to improve its use, regardless of the results of this exploratory and descriptive study on normal sonographic anatomy in a specific group of young adults, predominantly women and Caucasian.
Response #4: We appreciate the Reviewer’s comments and the opportunity to clarify the scope and intention of this study. The revised title (“…to guide the application of botulinum toxin”) does not imply that the present work offers a clinical protocol or injection algorithm; rather, it reflects that understanding normative anatomy is a prerequisite for any future guidance. If our aim had been to propose a clinical recommendation or injection protocol, the study would necessarily have been designed as a clinical trial with outcome-based evaluation, which is not the case here.
Regarding the remark that “variability exists” is already known, we agree that anatomical variability has been documented in earlier works and is clinically inferred from the occurrence of complications and suboptimal aesthetic outcomes. However, prior studies differ substantially in methodology and often examine only selected muscles. The primary objective of our study was therefore to establish a repeatable and standardized ultrasonographic protocol and to confirm, using high-resolution imaging, whether measurable variability can be consistently detected, quantified, and related to parameters previously described in heterogeneous literature. This step is essential before any clinically actionable guidance can be developed.
We acknowledge that this study does not define an injection algorithm. That was not its purpose. Rather, it represents the first necessary step: generating reliable normative sonographic values in healthy adults, demonstrating the feasibility of ultrasound for detailed assessment of upper-face musculature, and identifying the anatomical parameters that may warrant further exploration in clinical studies. The potential usefulness of ultrasound, and the justification for pursuing more advanced clinical research, has, in our view, been supported by the present findings.
Subsequent steps would require either:
(1) clinical outcome studies linking ultrasonographic parameters to treatment effects and safety (e.g., dose–effect relationships, spread patterns), or
(2) expanded exploratory work in broader age ranges, where ultrasound could quantify not only dimensions but also parameters such as echogenicity, elasticity, or microvascularity, all of which may reflect age-related muscular degeneration. Increasing heterogeneity with age provides new opportunities for descriptive characterization that would not be applicable in healthy young adults.
In this context, the normative values presented here do not lose their practical relevance. They serve as a baseline anatomical reference, enabling future studies to:
-compare older or pathological populations to a healthy standard,
-identify which anatomical parameters vary most strongly with age or clinical outcomes, and
-determine which measurements could eventually support a clinical injection algorithm.
Comment #5: Lines 294 to 358. The newly added section on clinical implications is very interesting but essentially remains speculative (based on previous literature on dosage in men) and is not directly derived from the data measured in this study.
Comment #5: We agree that the section on clinical implications contains elements that are inevitably speculative. This is inherent to the nature of the present study, which was designed as an exploratory, anatomical, ultrasound-based investigation, not a clinical trial. Without outcome data, it is not methodologically possible to draw definitive conclusions regarding dosing strategies or injection algorithms. As also noted in our response to Comment #1, establishing clinically actionable guidance would require full volumetric and geometric parameterization of individual muscles and a randomized comparison between landmark-based and ultrasound-guided injections, an approach fundamentally different from the objectives of this study.
The aim of the current work was considerably more foundational:
-to verify that the anatomical variability described in cadaveric studies can be consistently visualized and quantified on ultrasound,
-to present a scanning protocol and reproducible probe positions,
-to provide high-resolution representative ultrasound images for each muscle, and
-to demonstrate that ultrasound is a feasible tool for mapping the upper-face musculature which is treated with botox.
These elements constitute the necessary groundwork for future clinical research. Developing detailed recommendations regarding dosing or precise injection modifications for different anatomical variants would require a substantially larger sample and a dedicated clinical study with treatment outcomes.
Reviewer 2 Report
Comments and Suggestions for Authors
Authors revised manuscript according to my comments.
Paper can therefore be published without further changes.
Author Response
Thank you for reviewing our revised manuscript. We appreciate your helpful comments, which improved the paper, and are grateful for your time.
Reviewer 4 Report
Comments and Suggestions for Authors
The authors have dealt with the majority of the issues I raised. However, there are still 2 issues:
Lines 32-33 This is still incorrect. Toxin was not approved by FDA for forehead wrinkles in 2002. The approval was for treatment of glabellar lines only then. This must be changed.
Line 340 Whether the additional time taken for treatment using ultrasound is justified is a personal opinion of each treating clinician. In the authors opinion this is justified but many, many injectors will not have that view, especially if they are long-experienced in toxin treatment of the face. The authors should modify this part accordingly.
Author Response
Comment #1: Lines 32-33 This is still incorrect. Toxin was not approved by FDA for forehead wrinkles in 2002. The approval was for treatment of glabellar lines only then. This must be changed.
Response #1: We thank the Reviewer for this correction. The sentence has now been fully rewritten in the revised manuscript to reflect the accurate regulatory history.
Comment #2: Line 340 Whether the additional time taken for treatment using ultrasound is justified is a personal opinion of each treating clinician. In the authors opinion this is justified but many, many injectors will not have that view, especially if they are long-experienced in toxin treatment of the face. The authors should modify this part accordingly.
Response #2: We appreciate the Reviewer’s observation and agree that the perceived value of ultrasound guidance may vary among clinicians, particularly among highly experienced injectors. We have modified this section in the manuscript to acknowledge that the decision to incorporate ultrasound into routine aesthetic practice depends on individual clinical judgement, workflow preferences, and personal experience.